# Study on the Tribological Behavior and the Interaction between Friction and Oxidation of Graphite Reinforced by Impregnated Phosphate at High Temperatures

**DOI:** 10.3390/ma16093517

**Published:** 2023-05-04

**Authors:** Hao Cheng, Siyang Gao, Deli Duan, Shuai Yang, Weihai Xue, Bi Wu, Zhenguo Zhu

**Affiliations:** 1School of Materials Science and Engineering, University of Science and Technology of China, Shenyang 110016, China; 2Shi-Changxu Innovation Center for Advanced Materials, Institute of Metal Research, Chinese Academy of Sciences, 72 Wenhua Road, Shenyang 110016, China

**Keywords:** impregnated graphite, zinc phosphate, high-temperature, wear, oxidation, inert site

## Abstract

The stability of the graphite seal device is a key factor for the normal operation of aero engines. However, conventional graphite exhibits poor comprehensive performance due to its porous structure, which limits its application at high temperatures. Therefore, in this paper, phosphate was used to impregnated graphite pores, and the interaction between the friction, wear, and oxidation of phosphate-impregnated graphite against superalloy at high temperatures was studied through pin-on-disk friction tests. The results revealed that the coefficient of friction (COF) of matrix graphite fluctuated greatly, from 0.07 to 0.17, in the range of 100 °C to 500 °C, while the COF of impregnated graphite was stable, at around 0.13, from 100 °C to 500 °C. The wear rates of the two types of graphite were close from 20 °C to 300 °C, while the wear rate of the impregnated graphite was significantly lower than that of the matrix graphite at higher temperatures, from 400 °C and 500 °C. The reason was that the impregnated phosphate reacted with graphite at a high temperature, forming the inert site which helped to inhibit the oxidation and maintain the mechanical properties of the impregnated graphite at high temperatures. In addition, the impregnated graphite could maintain better integrity of the contact surface and reduce the inclusion of large hard metal oxides, thus effectively reducing the abrasive wear of the disk. Therefore, the wear depth of the superalloy disk samples with impregnated graphite was significantly lower than that of the matrix graphite. The results promote the application of phosphate-impregnated graphite under the high temperature conditions of aero engines.

## 1. Introduction

Due to a series of advantages such as low density, a low coefficient of thermal expansion, high thermal conductivity, self-lubrication, and chemical stability, graphite is a suitable sealing material, and the graphite seal is one of the most widely used forms of seals in aero engines [1,2,3,4,5]. However, with the continuous improvement of aero engines and the increase in inlet temperature and pressure [6,7], the service conditions of the crucial components, including the graphite seal, are deteriorating, which inevitably brings severe challenges to the tribological behavior and sealing performance of the graphite seal. Therefore, it is necessary to study the tribological behavior of graphite at high temperatures.

Early studies on the tribological behavior of graphite have focused on the influence of environmental factors, such as vacuum, inert gas, and water vapor, as well as the formation mechanism of friction-transferred film on the surface of the counterpart [8,9,10,11,12,13]. However, due to the high porosity, traditional graphite’s mechanical properties and resistance to high-temperature oxidation are poor, making it difficult to cope with the demanding operating conditions of advanced aero engines. Therefore, in recent years, the focus of research on graphite seals has gradually shifted to the preparation of reinforced graphite and the evaluation of its friction and wear behaviors. At present, the most common preparation process for reinforced graphite is vacuum impregnation, and typical reinforced graphite includes resin-reinforced graphite, inorganic salt-reinforced graphite, metal-reinforced graphite, etc. Moreover, the resin-reinforced graphite and metal-reinforced graphite exhibit higher strength and hardness, while the inorganic salt reinforced graphite shows higher resistance to high temperatures. Accordingly, the research on tribological behavior is mainly focused on these types of reinforced graphite [14,15].

Zhu [16] and Zhao [17,18], respectively, investigated the friction behavior of resin-impregnated graphite through pin-on-disk experiments. Zhu suggested that the impregnated graphite could form a more stable and complete friction-transferred film on the surface of the counterpart at high loads due to the interaction between the furan resin and the wear debris of the graphite, thus reducing the coefficient of friction (COF) and wear rate. Zhao concluded that the carbon-based friction-transferred film formed was more stable and homogeneous as the temperature increased, significantly improving the friction and oxidation resistance properties. Zhang [19,20,21] studied the friction behavior of resin-impregnated graphite under water and oil lubrication conditions, along with the wear mechanism between the resin-impregnated graphite and WC-Ni in sliding contact. Hirani [22] reported the ability of antimony-impregnated graphite to form a friction-transferred film on mating metal surfaces under dry, water, and steam environments. Wang [23] found that the homogeneous network metal distribution in the antimony/graphite composite contributed to the formation of a stable lubricating film on the surface of the counterpart, which helped to reduce the COF and the wear volume. Chen’s study showed that the main wear mechanism of phosphate-impregnated graphite under the oil lubrication condition was abrasive wear, and the removal of graphite was achieved through the fracture mechanism [24]. Qian [25] focused on the tribological properties of impregnated graphite in a corrosive environment. Tan [26], Wei [27], and Yang [28] studied the tribological behavior of Al-Fe-V-Si, Copper/Ti_2_SnC, and PTFE composites containing graphite at both room and high temperatures.

In summary, although there were a large number of studies on the tribological behaviors of impregnated graphite, and the wear mechanism was explored in depth, the experimental conditions were mostly limited to a relatively low temperature range, i.e., an oil lubrication condition where the temperature was usually not more than 200 °C. However, the current service environment temperature for graphite seals in advanced aero engines has exceeded 400 °C, and may even reach 500 °C, under certain extreme operating conditions. This would certainly demand high requirements on the high-temperature oxidation resistance and tribological properties of graphite sealing materials. The lack of research on high temperature wear mechanism and the interaction between oxidation and friction, however, limited the further application of impregnated graphite under extreme working conditions.

Based on these facts, in this paper, the phosphate-impregnated-reinforced graphite applied in engineering was selected as the research object, and the unimpregnated matrix graphite was selected as the reference object. The friction, wear, and oxidation behaviors of the two types of graphite at high temperatures were investigated through pin-on-disk and static oxidation tests. By means of scanning electron microscopy (SEM), energy spectrum analyzer (EDS), X-ray photoelectron spectroscopy (XPS), and Raman spectroscopy, the morphology and composition of the wear surface and friction-transferred film were characterized, and then the damage mechanism and the interaction between the friction and oxidation of graphite were deduced. The results of this paper provide important reference value for promoting the application of reinforced graphite under extreme working conditions.

## 2. Materials and Methods

### 2.1. Materials Preparation

The main production process of phosphate-impregnated graphite was as follows: The M265 was selected as the matrix graphite (Sinosteel Shanghai New Graphite Material Co., Ltd, Shanghai, China). Zinc phosphate hydrate (Zn_3_P_2_O_8_·xH_2_O) was dissolved in distilled water to prepare the solution. The matrix graphite was impregnated into the solution under 0.8 MPa for 5 h. Then, the sample obtained above was calcined at 800 °C for 2 h under argon atmosphere to obtain the phosphate-impregnated graphite.

To simulated the mating materials for graphite seal in practical engineering application, the nickel-based superalloy GH4169 (a Chinese alloy designation with a similar composition to Inconel 718) was selected as the friction pair material. The chemical compositions of the two types of the graphite and the GH4169 are shown in Table 1, and the physical and mechanical properties are shown in Table 2.

### 2.2. Friction and Oxidation Test

When the aero engine graphite seal device is under abnormal high temperature conditions, the lubricating oil would evaporate. Therefore, this research simplified the friction environment to an air atmosphere. The pin-on-disk tests were carried out by a multi-functional friction tester (CETR UMT 2, Bruker, Billerica, MA, USA) under dry friction conditions. The schematic diagram of the test apparatus and specimen size are shown in Figure 1.

Both types of graphite were machined into small cylindrical specimens, with ball heads as pin specimens. The GH4169 alloy was machined into a round sheet and used as the disk specimen. The pin and disk samples were abraded and polished with 80, 200, 400, 600, 800, 1000, 1200, and 1500 grit sandpapers and W2.5 water soluble diamond grinding paste using a polishing machine (SAPHIR 250 M1, Germany). Therefore, the surface roughness of the pin and disk samples was controlled to 0.1 ± 0.02 μm and 0.3 ± 0.02 μm, respectively. Before the test, all the specimens were ultrasonically cleaned in an alcohol bath for 10 min to remove the surface contaminants using a ultrasonic cleaner (Shanghai KUDOS Ultrasonic Instruments Co., Ltd., Shanghai, China).The pin-on-disk tests were performed at 20–500 °C. The resistance furnace of UMT 2 heated the friction ambient to the corresponding test temperature at a rate of 15 °C/min, which was maintained for 5 min to ensure that both upper/lower samples reach a stable temperature. The experiment parameters are listed in Table 3.

After the test, the depth of wear track on the surface of the disk was directly measured by the 3D optical profilometer (UP-Lambda2, Rtec, SAN Jose, CA, USA). The wear volume and wear rate of the graphite pin specimens were measured in the following manner. The volume of the spherical crown could be treated as the wear volume of the pin specimen, as shown in Figure 2. With the wear volume, the wear rate could be calculated through the following equations [29]:(1)$$\left.\begin{array}{c} V_{\mathrm{pin}}=\frac{\pi }{6}h\left(3h\left(2r-h\right)-h^{2}\right)\\ h=r-\sqrt{r^{2}-{\left(\frac{d}{2}\right)}^{2}} \end{array}\right\}$$
(2)$$\left.\begin{array}{c} W_{\mathrm{pin}}=\frac{V_{\mathrm{pin}}}{F_{n}\cdot s}\\ s=v\cdot t\\ v=2\cdot \pi \cdot R\cdot n/60 \end{array}\right\}$$
where $V_{\mathrm{pin}}$ was the wear volume of the graphite pin; mm^3^; *h* was the worn height of the ball head of the pin, which was equal to the depth of the spherical crown, mm; *d* was the diameter of the wear spot, mm; *r* was the original radius of the ball head of the pin, which was 2 mm; and $F_{n}$ is the normal load, which was 30 N. *R* was the radius of the wear track on the disk, which was 12.5 mm; *n* was the rotation speed, which was 60 r/min; *t* was the wear time, which was 7200 s; *v* was the sliding speed, which was 0.0785 m/s; and *s* was the sliding distance, which was 565.5 m.

It is well known that the oxidation of graphite at high temperatures leads to significant changes in its mechanical properties, which would, in turn, strongly affect the friction and wear behaviors. In order to clarify the effect of the oxidation on the friction and wear behaviors of the two types of graphite, static oxidation tests of air atmosphere were carried out using a muffle furnace (Zhengzhou Weida High-temperature Experimental Instruments Co., Ltd., Zhengzhou, China) at temperatures of 400 °C, 500 °C, and 600 °C, respectively. The specimen with a size of 5 × 5 × 5 mm was placed flat at the bottom of an alumina crucible to ensure that the surface and sides were in full contact with air. The duration of each static oxidation test was set at 2 h.

### 2.3. Characterization and Measurement

The polished surface morphology and elemental composition of the matrix graphite and the impregnated graphite were observed and measured using a TESCAN MIRA3 field emission scanning electron microscope (SEM, TESCAN Instruments, Czech Republic) equipped with an Ultim MaxN silicon drift-type energy spectrometer (EDS, Oxford Instruments, Abingdon, UK). D/mMax-2500PC-type X-ray diffraction (XRD, X’Pert PRO, Almelo, The Netherlands) was performed to investigate the phase composition and crystal structure of the two types of graphite. The scan was conducted using Cu Kα1 radiation (λ = 0.1541 nm) at the 2θ range from 10° to 90° at the rate of 4° per minute. The integrated patterns were then analyzed using Jade 6 for phase identification. The surface roughness of the alloy specimen was measured with the Profilometer 2300A-R (Genertec Harbin Measuring & Cutting Tool Co., LTD, Harbin, China) prior to the friction tests. After the friction tests, the sectional profile in the wear track of the counterpart disk was measured with a 3D optical profilometer (UP-Lambda2, Rtec, Silicon Valley, CA, USA). The wear surface morphology of the graphite and alloy disk and the state of the graphite after static oxidation were obtained using an optical digital microscope (VHX-6000, Keyence, Osaka, Japan) and SEM. The structure of wear surface and the friction-transferred film on the counterpart surface of the two types of graphite were detected by Raman spectroscopy (Lab-Ram HR Evolution, Horiba, Kyoto, Japan) using 532 nm as the excitation laser wavelength. The chemical composition and state of friction–oxidation products on the wear tracks of the counterpart were analyzed by X-ray photoelectron spectroscopy (XPS, ESCALAB 250, ThermoVG, Waltham, MA, USA). All the peaks were calibrated using the C 1 s peak and fitted using Thermo Advantage 5.52 software.

## 3. Results

### 3.1. Microstructure and Phase Composition of the Graphite

Figure 3a–c shows the surface SEM morphology and EDS surface mapping. Figure 3d is the XRD pattern of the impregnated graphite and the matrix graphite. It can be observed that the matrix graphite is mainly composed of a dark color graphite phase and a certain amount of black-colored pores. In contrast, the impregnated graphite contains a certain amount of bright white color phosphate phase, in addition to the graphite phase and the pores. Obviously, the pore content in the impregnated graphite is significantly lower than that in the matrix graphite, which can mainly be attributed to the filling effect of phosphate on the pores.

### 3.2. Tribological Properties 

The real-time COF curves for the two types of graphite at 20 °C, 300 °C, and 500 °C are shown in Figure 4a–c, and the summary results of the COF are shown in Figure 4d. At 20 °C, the COF curves of the two types of graphite are very smooth. The COFs of the matrix graphite and impregnated graphite are stable at around 0.22 and 0.20 after the run-in period, respectively. At 300 °C, the COF curve of the matrix graphite decreases first, and then increases, and finally stabilizes at 0.18, while the COF of the impregnated graphite shows a rapid increase at the beginning, and stabilizes at around 0.16 later on. At 500 °C, the COF curves of the two types of graphite rise rapidly and stabilize at 0.14 and 0.16, respectively. Figure 4d shows the average COFs of the two types of graphite at different temperatures. Comparing the COFs at all temperatures, the COF of the impregnated graphite stabilizes around 0.13 from 100 °C to 500 °C, while the COF of the matrix graphite fluctuates sharply, although it is only 0.075 at 200 °C.

Figure 5 shows the wear rate of the two types of graphite at different temperatures. At low temperatures (20 °C, 100 °C, and 200 °C), the wear rates of the both are relatively small and close to each other. Beginning from 300 °C, the wear rate of the matrix graphite increases significantly with the increase in temperature. Its wear rate at 500 °C is nearly 24 times higher than at room temperature. In contrast, the increase in the wear rate for impregnated graphite is relatively moderate with the increase in temperature. The wear rate of the impregnated graphite at 500 °C is only 8 times that at 20 °C. In addition, a direct comparison between the two types of graphite shows that the wear rate of impregnated graphite is only one-third of that of the matrix graphite at 500 °C.

Figure 6 shows the 3D and local SEM wear morphology images of the two types of graphite at different temperatures. At 20 °C, the wear of the matrix graphite and the impregnated graphite is relatively slight, and the worn surface is flat, dominated by wrinkles. With the increase in temperature, the proportion of furrow wear increases. Compared to the matrix graphite, the groove-like damage caused by abrasive wear on impregnated graphite is relatively minor, with surface-localized bulge being the main wear morphological feature. At the same time, there are some wrinkles, spalling, and wear particles of different sizes on the wear scar of both of the two types of graphite. At 500 °C, the number of grooves on the wear scar of the matrix graphite decreases and both the two graphite’s wear morphology are similar, showing slight grooves and spalling pits.

Figure 7 shows the worn surface morphology and the profile curves of the wear track (along the radius of the wear track) of the disk specimens rubbed against the two types of graphite at different temperatures. At 20 °C, the worn surface is very smooth, and there is almost no obvious fluctuation in the profile curve, indicating that the wear of the disk is very slight and approximately negligible. From the beginning at 100 °C, furrow and friction-transferred film appear on the surface, and the relatively amount of the friction-transferred film increases significantly with the increase in temperature. In addition, the fluctuation of the profile curve intensifies significantly with the increase in temperature, which means the wear of the disk is aggravated. Comparing the two types of disk specimens, it is found that the maximum depth of the wear track on the disk specimens paired with the impregnated graphite is significantly smaller than that on the disk paired with the matrix graphite under this condition, as shown in Figure 8.

### 3.3. Structure and Composition of the Graphite Friction-Transferred Film

Figure 9 shows the Raman spectra of the wear surface and the friction-transferred film of the two types of graphite. Generally, the carbon material has three main characteristic peaks: the D peak (1350 cm^−1^), the G peak (1580 cm^−1^), and the 2D peak (2710 cm^−1^), respectively. The G peak corresponds to the ideal graphite vibrational mode with E2g symmetry, and the 2D peak corresponds to the stacking characteristics of graphite, both of which reflect the ordered structure of graphite [24,30,31]. The D peak is recognized as a defective peak, reflecting the disordered nature of the graphite structure. The intensity ratio I_D_/I_G_ grows with increasing disorder in the graphitic structure.

From 20 °C to 300 °C, the I_D_/I_G_ values of the wear surfaces of the two types of graphite decrease significantly with the increase in temperature, indicating that high temperature friction strongly increases the degree of the wear surface structure, as shown in Figure 9a,b. The I_D_/I_G_ values of the friction-transferred film on the disk specimen follows a trend similar to that of the graphite’s wear surface, as shown in Figure 9c,d. However, the difference is that the characteristic peaks on the disk specimen paired with the matrix graphite disappear at 500 °C, which means the graphitization declines. The evolution of the Raman characteristic peaks shows that, under the influence of friction shear stress and thermal stress, the structure of the wear surface and the friction-transferred film on the disk specimens of the two types of graphite tend to change from disordered to ordered with the increase in temperature. Consequently, the graphite wear surface and the friction-transferred film could slide more easily relative to each other [32], which is helpful for the reduction of the COF.

Figure 10 shows the O 1s fine spectra of the friction-transferred film on the disk specimen. For the disk paired with matrix graphite, the dominant chemical bond is C=O at 200 °C, 400 °C, and 500 °C, with a peak position of 531.5 eV [31,33,34]. For the disk paired with impregnated graphite, there is an extra C-O-P bond (532.9 eV), in addition to C=O bond (531.5 eV) [35,36,37], at 400 °C and 500 °C, which indicates that a high-temperature friction-induced chemical reaction between the phosphate and graphite occurred on the contact interface, in addition to the oxidation of graphite.

Figure 11 shows the full XPS spectra at 500 °C and the fine spectra of the Ni 2p, Cr 2p, and Fe 2p of the wear tracks of the counterpart disks at 500 °C. Figure 11a shows that the predominant elements on the wear tracks are C and O elements, with a small amount of metallic elements being oxidized. The Ni 2p spectrum could be decomposed into three main peaks at 857.2 eV, 861.4 eV, and 873.8 eV, all of which are NiO [38,39,40]. The fine spectrum of Cr 2p shows that the oxide is mainly Cr_2_O_3_, with peaks at 576.2 eV and 586 eV, and a small amount of CrO_3_, with a peak at 579.8 eV [39,41,42]. The fine spectrum of Fe 2p has three main fitted peaks at 710.7 eV, 713.3 eV, and 725.2 eV, respectively, with the corresponding oxides being Fe_2_O_3_ [40,43,44,45]. The results of XPS show that the primary friction oxidation products are mainly composed of NiO, Cr_2_O_3_, and Fe_2_O_3_, which are mixed in the friction-transferred film.

### 3.4. Static Oxidation of the Graphite

The weight loss results obtained are shown in Figure 12. Compared to the oxidation weight loss of the matrix graphite, which increases dramatically with the increase in temperature, the oxidation weight loss of the impregnated graphite at different temperatures is very small and relatively close, indicating an excellent and stable oxidation resistance for the impregnated graphite, especially at high temperatures.

Figure 13 shows the surface SEM morphology and EDS mapping of the two types of graphite after static oxidation at 600 °C. For the matrix graphite, serious spalling and large sized pores (40 μm–60 μm) appeared on the surface, indicating a significant oxidation. In contrast, the surface integrity of the impregnated graphite is maintained quite well, and the phosphate is evenly dispersed on the surface. The uniform dispersion of phosphate, on the one hand, is helpful to reduce the amount and size of pores (1 μm–5 μm); on the other hand, it can effectively block the direct contact between oxygen and the active point of graphite. As a result, the oxidation of the impregnated graphite is effectively inhibited, and its mechanical properties are well maintained [12,46,47,48].

## 4. Discussion

By a comprehensive comparison of the test results of the COF, the wear rate of the two types of graphite, and the damage of the counterpart disk (wear track depth), it can be seen that the impregnated graphite is superior to the matrix graphite in all aspects, especially in its own wear resistance and wear reduction of the counterpart disk at high temperatures. Therefore, it is necessary to carry out a further in-depth discussion regarding the above two aspects.

### 4.1. Mechanism for Resisting Wear of the Impregnated Graphite

For the impregnated graphite and matrix graphite, at room temperature (20 °C), the order degree of the wear surface structure is low (higher I_D_/I_G_ value), and no obvious friction-transferred film is formed on the surface of the counterpart disk; therefore, the friction reduction effect at 20 °C is poor, and the COF is the highest. Due to the lower temperature, however, the mechanical properties could be well maintained for both the graphites. Therefore, despite the highest COF and the maximum friction force at room temperature, the two types of graphite can still maintain good wear resistance, showing the lowest wear rate.

As the temperature increases (100–200 °C), the order degree of the wear surface and friction-transferred film rises significantly (I_D_/I_G_ shows a decreasing trend with the increase in temperature), resulting in the decrease in COF and frictional shear stress, which is conducive to reducing the wear of the two types of graphite. However, the mechanical properties (hardness, strength, etc.) of the wear surface of the two types of graphite decreases, which promotes wear. The combined effect of the reduced COF and the reduced mechanical properties results in no significant change in the wear rate compared to the effect at room temperature (20 °C).

When the temperature rises to 300 °C, although the order degree on the surface of the two types of graphite and the friction-transferred film of the counterpart disk could still well maintain a lower I_D_/I_G_ value, the high-temperature oxidation (despite the experimental temperature of 300 °C, the friction heat also leads to high temperature, and the coupling of the two may cause the temperature of the friction interface to reach or even exceed 400 °C) would cause a further decrease in mechanical properties; therefore, the wear rate increases significantly with the increase in temperature compared to that at low temperatures (20–200 °C).

At 400 °C and 500 °C, for the matrix graphite, the oxidation led to further degradation of its mechanical properties and serious damage to its surface integrity (large amount of spalling and large size pores). Therefore, the wear rate increases sharply and reaches about 24 times that at room temperature. For the impregnated graphite, due to the presence of phosphate, which reacted with the graphite to form C-O-P inert sites, high temperature oxidation is effectively inhibited, and the high temperature mechanical properties are better maintained. In addition, the friction-transferred film on the surface of the disk paired with the impregnated graphite (500 °C) could maintain a good degree of order, while the order degree in the friction transfer film on the disk samples paired with the matrix graphite almost disappeared (for comparison, the friction-transfer film of the matrix graphite was severely degraded). As a result, the wear of the impregnated graphite at high temperatures (400 °C and 500 °C) is significantly lower than that of the matrix graphite.

### 4.2. Mechanism for Reducing the Wear of the Counterpart of the Impregnated Graphite

Since phosphate could fill the pore and effectively reduce the porosity, the impregnated graphite has excellent oxidation resistance and therefore, could well maintain mechanical properties at high temperatures. In contrast, the graphite matrix with many pores is oxidized significantly at high temperatures, and its mechanical properties become worse. Therefore, the integrity of the contact surface of the matrix graphite is worse than that of the impregnated graphite, which is characterized by spalling pits and more large-sized pores, resulting in more hard metal oxides accumulated on the surface of the matrix graphite.

The wear surface of the matrix graphite embedded hard metal oxide particles with larger volume and irregular shape from the disk at different temperatures, especially at 500 °C, as shown in Figure 14. These metal oxide particles would increase the degree of two-body or three-body abrasive wear during the friction process. Therefore, the depth of the wear tracks on the disks rubbed against the matrix graphite is larger than those rubbed against the impregnated graphite at the same temperature, as shown in Figure 15a–d. The distribution of P and C elements also agrees with the XPS results showing the formation of C-O-P inert sites on the friction-transferred film at 500 °C, as shown in Figure 15d.

## 5. Conclusions

When the graphite seal device in an aero engine is in the dry friction condition, with no oil, the temperature between the friction interface of the seal pair graphite/GH4169 will rise sharply, thus increasing the wear and affecting the normal operation of the engine. In order to better understand the tribological performance of zinc phosphate-impregnated graphite under such harsh conditions, we studied the friction and wear behavior of phosphate-impregnated graphite at high temperatures, the effects of the phosphate-impregnated components on the graphite wear, oxidation, and the structural evolution of friction-transferred film were investigated. In this study, pin-on-disk friction experiments with impregnated graphite and matrix graphite against GH4169 were carried out. SEM, the white light interferometer, Raman, and XPS were used to analyze graphite wear surfaces, graphite friction-transferred film on the alloy wear tracks, and the oxidation of the two graphites at high temperatures to summarize the corresponding interaction mechanism of the friction, wear, and oxidation. The conclusions are as follows:The COFs of the matrix graphite and the impregnated graphite do not differ much. The reduction in pores after impregnation with phosphate makes the COF of the impregnated graphite more stable at different temperatures (100–500 °C).At high temperatures (300–500 °C), the wear rate of the matrix graphite increases significantly under the combined effect of friction and oxidation, and the wear rate at 500 °C is 24 times that at room temperature (20 °C).At high temperatures (400–500 °C), the inert sites formed by the phosphate on the friction-transferred film of the impregnated graphite effectively inhibit the oxidation, slowing down the destruction of the mechanical properties of the impregnated graphite and thus reducing the wear rate.The impregnated graphite is reinforced with phosphate, which provides it with less porosity and a smoother friction surface at high temperatures. Therefore, the furrow damage of the impregnated graphite counterpart is lighter than that of the matrix graphite counterpart.

The results of this study confirmed that phosphate-impregnated graphite has a certain wear-resisting and friction-reducing effect under dry friction, without oil, at high temperatures in aero engines, which is helpful for maintaining the working stability of graphite seals. However, the work also has some limitations. In the future, we will conduct further research under high temperatures, such as introducing lubricating oil into the friction interface and studying the influence of a lubricating medium on phosphate-impregnated graphite tribological behavior.

## Figures and Tables

**Figure 1 materials-16-03517-f001:**
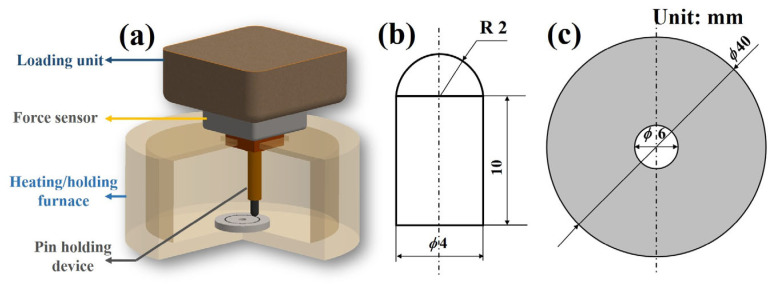
(**a**) Schematic diagram of high-temperature friction test apparatus, (**b**) geometry of graphite pin specimen, and (**c**) geometry of alloy disk specimen.

**Figure 2 materials-16-03517-f002:**
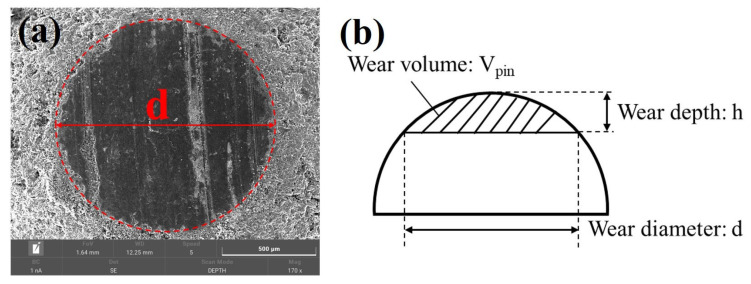
(**a**) Representative wear scar on the ball of the pin specimen observed via SEM; (**b**) schematic of the wear volume of the ball head.

**Figure 3 materials-16-03517-f003:**
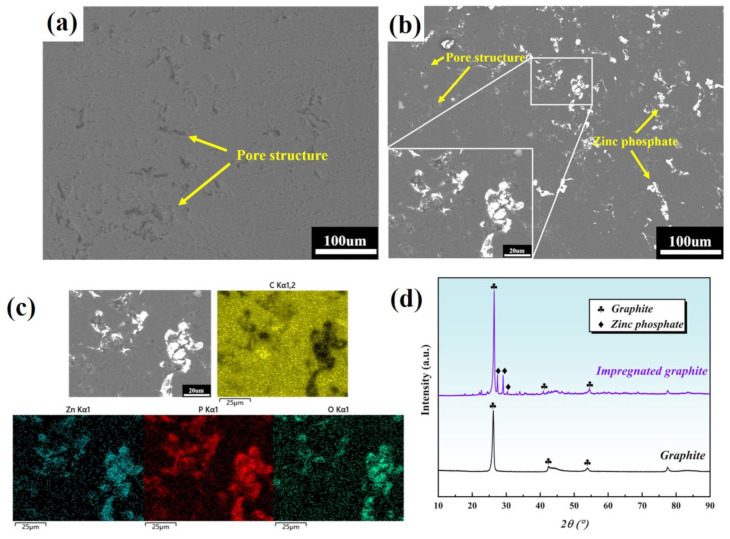
Characterization of surface microstructure: (**a**) SEM image of the matrix graphite; (**b**) SEM image of the impregnated graphite; (**c**) EDS mapping of the impregnated graphite; (**d**) X-ray diffraction pattern of the matrix graphite and the impregnated graphite.

**Figure 4 materials-16-03517-f004:**
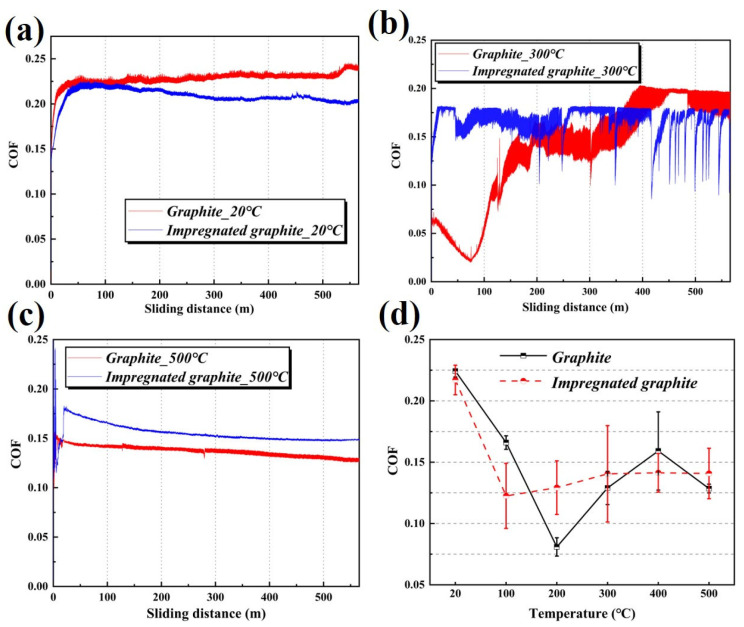
COF as a function of the sliding distance of matrix graphite and impregnated graphite at different temperatures of (**a**) 20 °C, (**b**) 300 °C, and (**c**) 500 °C, and (**d**): the average COF from 20 °C to 500 °C.

**Figure 5 materials-16-03517-f005:**
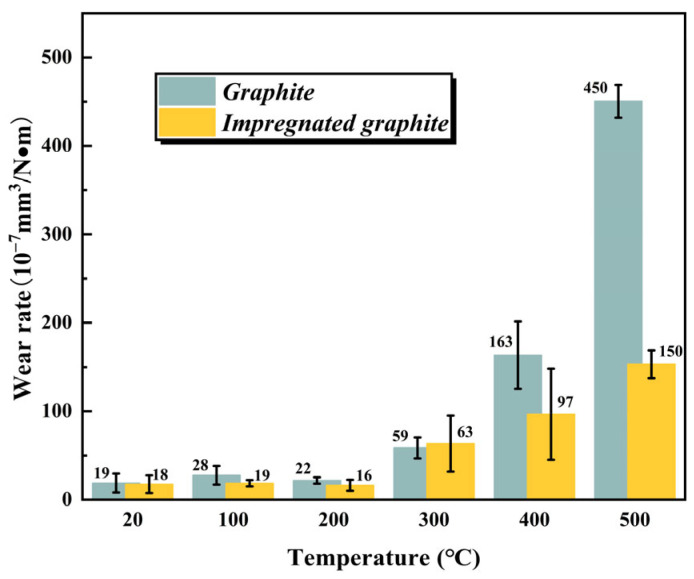
The wear rate of the matrix graphite and the impregnated graphite at different temperatures.

**Figure 6 materials-16-03517-f006:**
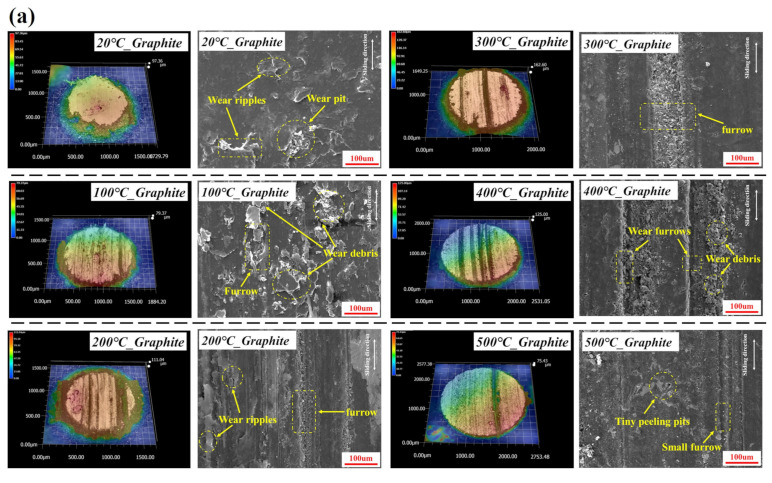
3D topography and SEM images of the wear surface on (**a**) matrix graphite and (**b**) impregnated graphite at different temperatures.

**Figure 7 materials-16-03517-f007:**
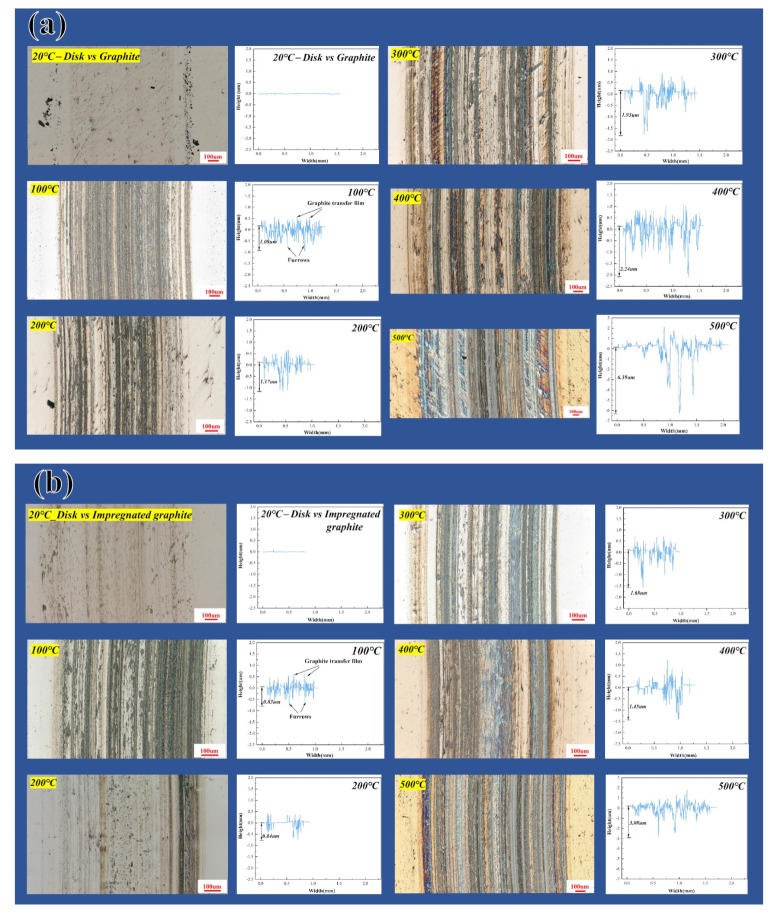
Wear track morphology and profile curves of wear track on the disk rubbed against (**a**) matrix graphite and (**b**) impregnated graphite under different temperatures.

**Figure 8 materials-16-03517-f008:**
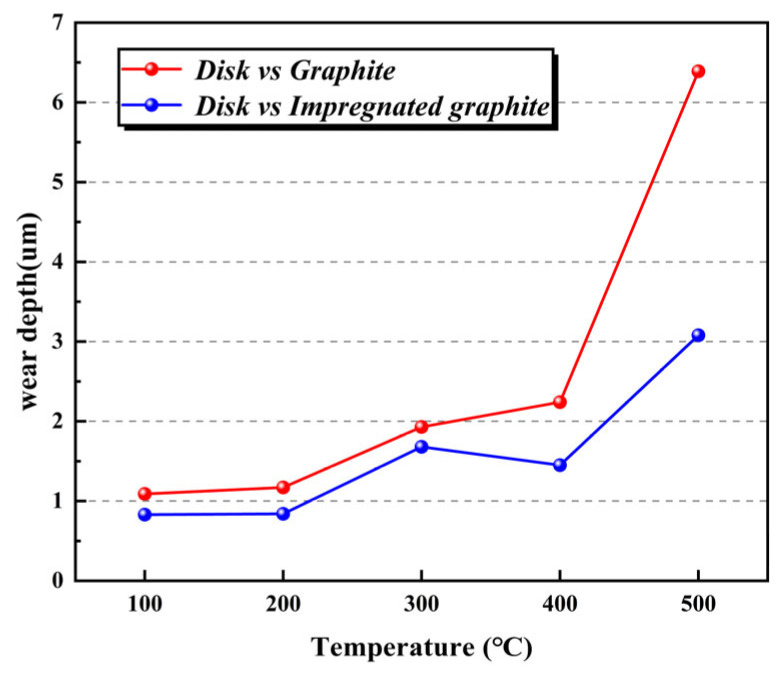
The maximum depth of the wear track of the alloy disk rubbed against matrix graphite and impregnated graphite under different temperatures.

**Figure 9 materials-16-03517-f009:**
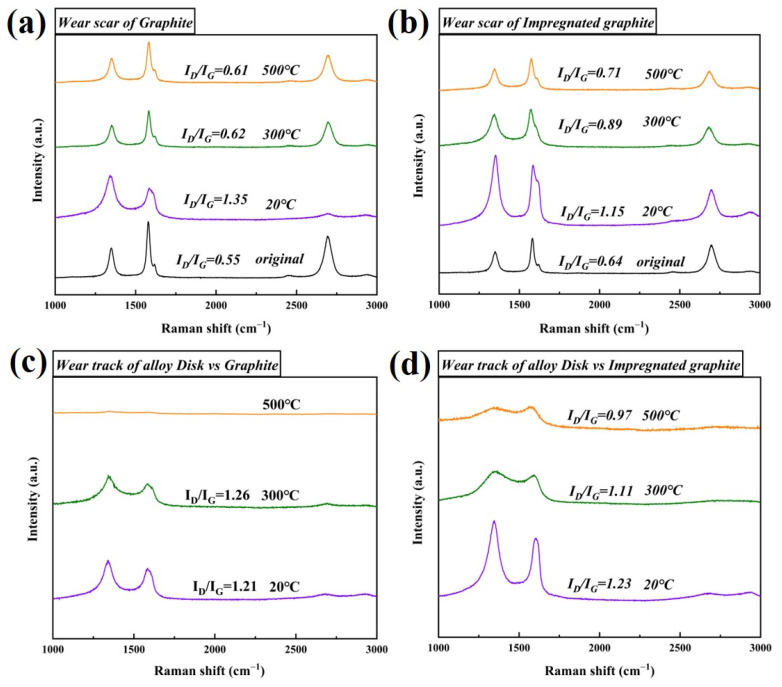
Raman spectra of the graphite wear surface and the disk wear track at different temperatures: (**a**) matrix graphite; (**b**) impregnated graphite; (**c**) disk vs. matrix graphite; (**d**) disk vs. impregnated graphite.

**Figure 10 materials-16-03517-f010:**
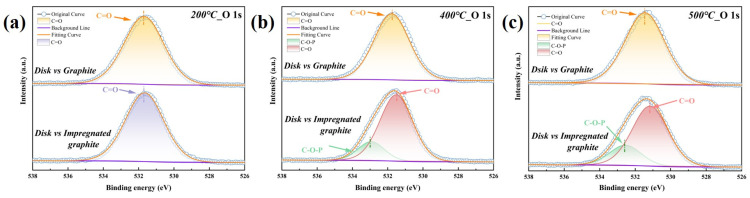
XPS high-resolution spectra of O 1s on the alloy wear tracks at different temperatures: (**a**) 200 °C, (**b**) 400 °C, and (**c**) 500 °C.

**Figure 11 materials-16-03517-f011:**
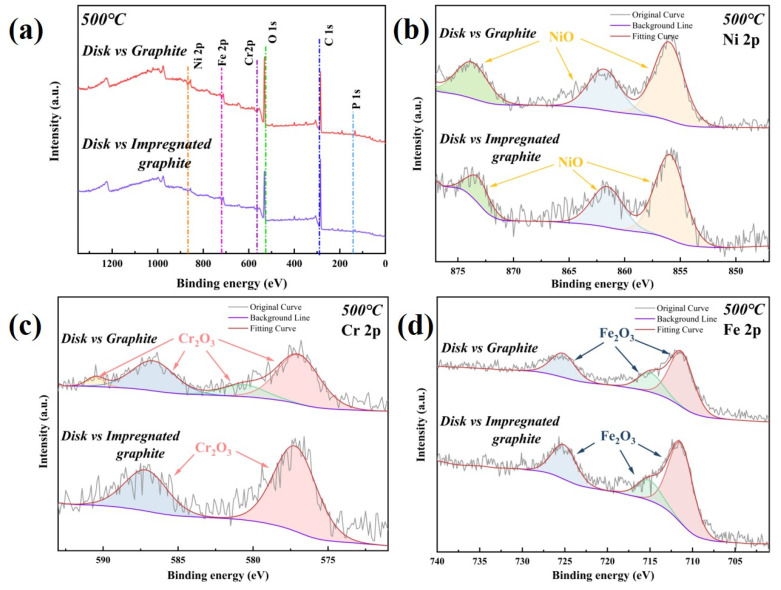
XPS spectra of high-temperature alloy wear surface at 500 °C: (**a**) survey spectrum; (**b**) Ni2p; (**c**) Fe2p; and (**d**) Cr2p.

**Figure 12 materials-16-03517-f012:**
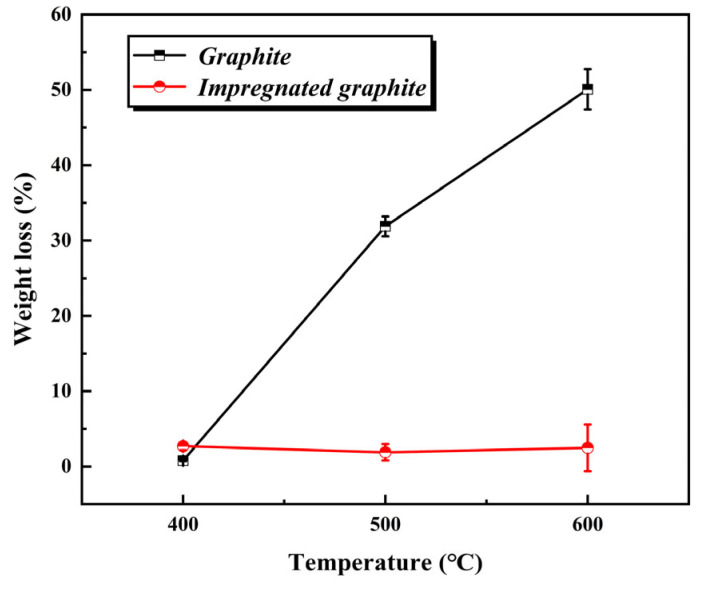
Static oxidation weight loss of the matrix graphite and impregnated graphite at 400 °C, 500 °C, and 600 °C.

**Figure 13 materials-16-03517-f013:**
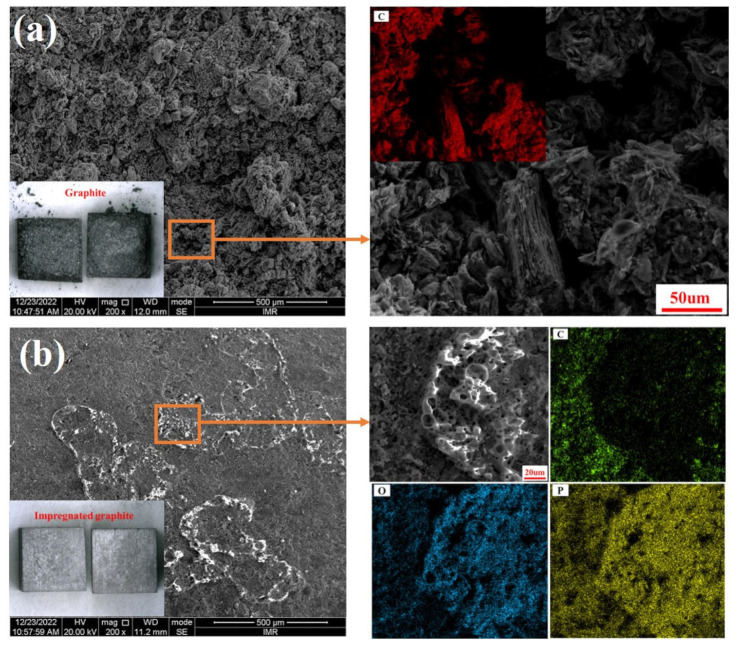
SEM images and EDS maps showing the oxidation morphology and element distribution of (**a**) matrix graphite and (**b**) impregnated graphite at 600 °C.

**Figure 14 materials-16-03517-f014:**
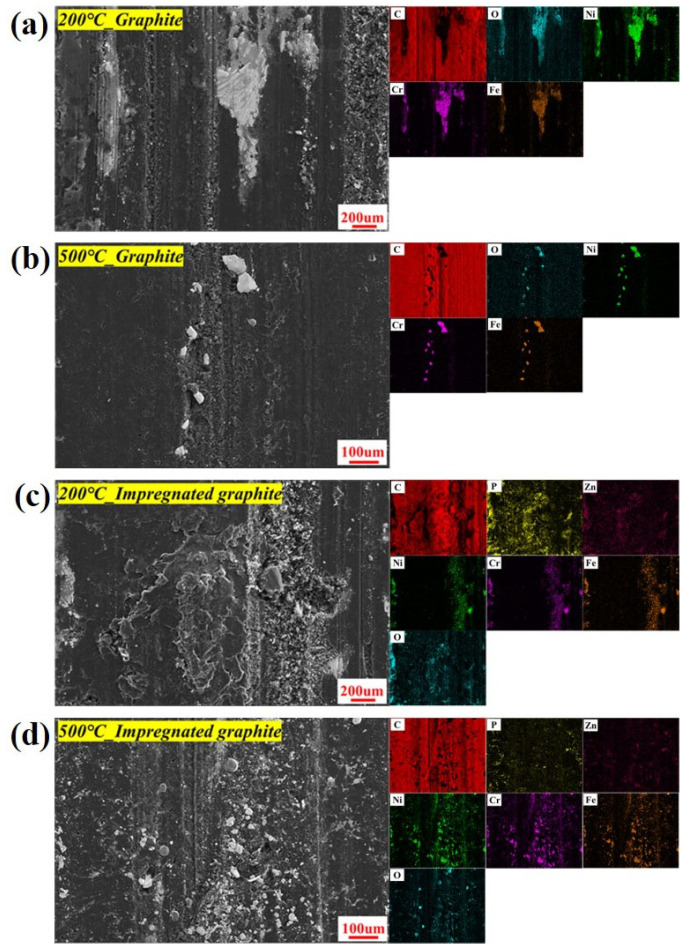
SEM micrograph and elemental distribution on the wear surface of the two types of graphite: (**a**) 200 °C, matrix graphite; (**b**) 500 °C, matrix graphite; (**c**) 200 °C, impregnated graphite; (**d**) 500 °C, impregnated graphite.

**Figure 15 materials-16-03517-f015:**
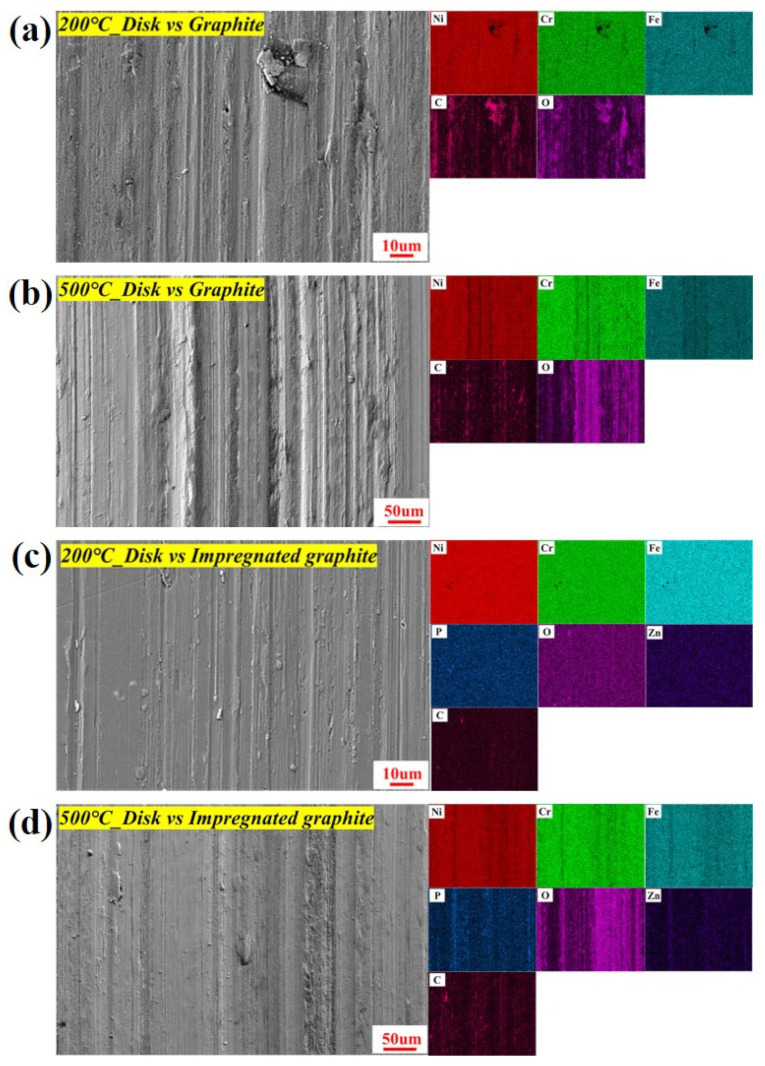
SEM micrograph and elemental distribution on the transfer film of two types of graphite: (**a**) 200 °C, disk vs. matrix graphite; (**b**) 500 °C, disk vs. matrix graphite; (**c**) 200 °C, disk vs. impregnated graphite; (**d**) 500 °C, disk vs. impregnated graphite.

**Table 1 materials-16-03517-t001:** Chemical compositions of the matrix graphite, impregnated graphite, and GH4169 (wt.%).

Composition	C	O	P	Zn	Cr	Ti	Al	Mo	Ni	Nb	Fe
Matrix graphite	100	-	-	-	-	-	-	-	-	-	-
Impregnated graphite	87~91	7~9	1~2	1~2							
GH4169	0.04	-	-	-	18.56	1.02	0.57	3.06	52.89	5.10	Balance

**Table 2 materials-16-03517-t002:** Physical and mechanical parameters of the two types of the graphite and the GH4169.

Specimen	Density (g/cm^3^)	Compressive/Tensile Strength (MPa)	Flexural Strength (MPa)	Hardness	Open Porosity (%)
Matrix graphite	1.88	105	41	75 HS	9.0
Impregnated graphite	1.96	112	44	77 HS	3.0
GH4169	8.24	965	-	380 HBS	-

**Table 3 materials-16-03517-t003:** Experimental parameters of the high temperature pin-on-disk test.

Parameters	Value
Pin	Phosphate-impregnated graphite Matrix graphite
Disk	GH4169
Temperature (°C)	20, 100, 200, 300, 400, 500
Time (s)	7200
Load (N)	30
Rotation speed (rpm/min)	60
Rotation radius (mm)	12.5

## Data Availability

The data presented in this study are available upon request from the corresponding author.

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
