# Peer review of "Study on the Tribological Behavior and the Interaction between Friction and Oxidation of Graphite Reinforced by Impregnated Phosphate at High Temperatures"

_materials, 2023, doi:10.3390/ma16093517_

Round 1
Reviewer 1 Report
Graphite materials have been widely used as ideal friction materials in various equipment, primarily due to the fact that and outstanding tribological properties. In general, graphite contains many pores, hence, to improve its properties, it is usually impregnated with metals or resins that fill its open pores. In order to study the friction and wear behavior of phosphate-impregnated graphite at high temperature, the authors of this manuscript were the effects of the phosphate-impregnated components on the graph- ite wear, oxidation and the structural evolution of friction-transferred film. The results provide an important experimental basis for designing high-property graphite-based friction materials in industrial applications. The article is relevant and has scientific value.
The authors should pay attention to the following remarks and recommendations.
1. Authors need to improve abstract. At the beginning of the abstract, you need to add a Background: Place the question addressed in a broad context and highlight the purpose of the study. At the end of the abstract, add conclusions or interpretations of an applied nature.
2. Please check the traditional names for research manuscript sections are correct. In particular, section 2 should be entitled "Materials and Methods".
3. I would like to remind you that research methods should be described with sufficient detail to allow others to replicate and build on published results. In the text, it is necessary to indicate by which method and with the help of which devices the roughness of the pins and disks was determined. It is also necessary to specify which ultrasonic device the authors used to clean the samples. How temperature was maintained during the pin-on-disk tests? It is not clear to me why the size of the disk in Fig. 1, c does not correspond to the size indicated in Table 3.
4. There is probably an error in the second expression of system (1), line 124.
5. Recommendation for improvement of conclusions. The findings and their implications should be discussed in the broadest context possible, and also indicate the applied value of the research. It is also necessary to indicate the limitations of the results or methods obtained and outline the direction of further research.

Reviewer 2 Report
In my opinion, the article is very interesting. The topic of the article is very interesting. I believe that it has been fully described and clearly discussed. I recommend the above article to be published in Materials.
Author Response
Thank you very much for your support of our research.
Reviewer 3 Report
The article is interesting and raises important issues of the influence of the structure of graphite seals on their tribological properties. In the introduction, the authors relate these issues to aircraft engines. Hence, there were some ambiguities in the research methodology, which, in my opinion, should be clarified, and perhaps the research should also be supplemented.
1. The subject of the research is the tribological result of graphite impregnation with a phosphate solution. The authors hardly describe how graphite is impregnated, and the term "phosphate solution" is imprecise. I believe that the method of impregnation should be described in detail.
2. In the introduction, the authors pointed to various known methods of graffiti impregnation. In my opinion, it is necessary to compare the results obtained by the authors with the results obtained for other impregnants.
3. The basic test is the study of the coefficient of friction and wear. My guess is that dry friction (technically dry) was used in the study. The authors did not specify the composition of the atmosphere in which the tribological process was carried out. At high temperature, an oxidation process also takes place, which modifies the surface layer of the ball and disc. The course of this process depends on the composition of the environment (air, vapors of organic compounds). The authors described the results of surface analysis after friction, but they did not relate the test conditions to the actual working conditions of graphite seals, e.g. in aircraft engines. Whether these seals operate in an air atmosphere or come into contact with other substances from lubricating oil, fuel or exhaust gases. This should be explained in the description of the research methodology, and then taken into account when formulating conclusions.
Round 2
Reviewer 3 Report
The changes made the article much clearer. Thank you for considering my suggestions.